# Therapeutic Vulnerabilities in *FLT3*-Mutant AML Unmasked by Palbociclib

**DOI:** 10.3390/ijms19123987

**Published:** 2018-12-11

**Authors:** Iris Z. Uras, Barbara Maurer, Sofie Nebenfuehr, Markus Zojer, Peter Valent, Veronika Sexl

**Affiliations:** 1Institute of Pharmacology and Toxicology, University of Veterinary Medicine, 1210 Vienna, Austria; iris.uras@vetmeduni.ac.at (I.Z.U.); barbara.maurer@vetmeduni.ac.at (B.M.); sofie.nebenfuehr@vetmeduni.ac.at (S.N.); markus.zojer@vetmeduni.ac.at (M.Z.); 2Ludwig Boltzmann Institute for Hematology & Oncology, Medical University of Vienna, 1090 Vienna, Austria; peter.valent@meduniwien.ac.at; 3Department of Internal Medicine I, Division of Hematology and Hemostaseology, Comprehensive Cancer Center, Medical University of Vienna, 1090 Vienna, Austria

**Keywords:** CDK6, palbociclib, AURORA kinases, AKT, *FLT3–ITD*, *FLT3*–D835Y, AML

## Abstract

While significant progress has been made in the treatment of acute myeloid leukemia (AML), not all patients can be cured. Mutated in about 1/3 of de novo AML, the FLT3 receptor tyrosine kinase is an attractive target for drug development, activating mutations of the *FLT3* map to the juxtamembrane domain (internal tandem duplications, ITD) or the tyrosine kinase domain (TKD), most frequently at codon D835. While small molecule tyrosine kinase inhibitors (TKI) effectively target ITD mutant forms, those on the TKD are not responsive. Moreover, FLT3 inhibition fails to induce a persistent response in patients due to mutational resistance. More potent compounds with broader inhibitory effects on multiple *FLT3* mutations are highly desirable. We describe a critical role of CDK6 in the survival of *FLT3*^+^ AML cells as palbociclib induced apoptosis not only in *FLT3–ITD*^+^ cells but also in *FLT3*–D835Y^+^ cells. Antineoplastic effects were also seen in primary patient-derived cells and in a xenograft model, where therapy effectively suppressed tumor formation in vivo at clinically relevant concentrations. In cells with *FLT3*–ITD or -TKD mutations, the CDK6 protein not only affects cell cycle progression but also transcriptionally regulates oncogenic kinases mediating intrinsic drug resistance, including AURORA and AKT—a feature not shared by its homolog CDK4. While AKT and AURORA kinase inhibitors have significant therapeutic potential in AML, single agent activity has not been proven overly effective. We describe synergistic combination effects when applying these drugs together with palbociclib which could be readily translated to patients with AML bearing *FLT3–ITD* or –TKD mutations. Targeting synergistically acting vulnerabilities, with CDK6 being the common denominator, may represent a promising strategy to improve AML patient responses and to reduce the incidence of selection of resistance-inducing mutations.

## 1. Introduction

Acute myeloid leukemia (AML) is the most common form of acute leukemia in adolescents and adults. AML is a malignant disorder of hematopoietic cells characterized by an accumulation of proliferating blast cells blocked in differentiation at various stages. Although significant progress has been made in treating many types of leukemias during recent years, AML remains a deadly disease in many cases, with survival rates lagging behind other blood cell cancers. Long-term remission is only achieved in a minority of all patients, the disease recurs frequently after therapy, and many patients ultimately die of their disease or from the toxicity of repeated chemotherapy and/or stem cell transplantation [1].

The intensive efforts to define the pathogenesis of AML have enabled the development of novel therapeutic drugs targeting key molecular players. Mutated in about 30% of AML, Fms-like tyrosine kinase 3 (FLT3) represents one of the most attractive targets. Two distinct *FLT3* mutations occur at different hotspots: internal tandem duplications in the juxtamembrane domain (*FLT3*–ITD) and point mutations in the tyrosine kinase domain (*FLT3*–TKD), most commonly at codon D835. A stable expression of activating *FLT3* mutations in 32D and Ba/F3 cells confers cytokine independency. *FLT3*–TKD mutations induce the activation of the RAS, ERK, and AKT pathways, which is similar to what occurs with *FLT3*–ITD mutations, but show relatively weak STAT5 activity [2]. These differences in signaling may help to explain distinct disease courses and progression patterns and thus a different prognosis: while *FLT3*–ITD mutations are associated with an aggressive disease course and are strong predictors of rapid relapse and short overall survival (OS) after chemotherapy, AML patients with *FLT3*–TKD mutations develop a less-aggressive disease with a somehow better prognosis.

To date, several small molecule tyrosine kinase inhibitors of FLT3 (FLT3–TKI) have been developed [3,4]. Their clinical efficacy is limited due to a lack of potency and development of drug resistance: the response to monotherapy is typically short-lived even when TKI is given continuously, rendering it difficult to bridge patients to allogeneic stem cell transplantation. Existing TKI exhibit a different spectrum of activities against *FLT3* alterations: FLT3 kinase domain mutants, such as D835, are inherently resistant to FLT3 inhibitors [5,6]. The detection of mutations within *FLT3*–TKD only recently emerged as an important mechanism of therapy resistance [7]. Such mutations are found on *FLT3*–ITD alleles in patients relapsing from TKI therapy and induce amino acid substitutions within the kinase domain, generally at position D835 (D835Y/V/F) [8,9,10]. These double-mutant leukemic clones are not detected at the time of diagnosis but may be provoked and expand by the selective pressure delivered by TKI treatment. The prognosis of relapsing patients is poor and demands alternative strategies to prevent and attack the deadly *FLT3*-mutated leukemic blasts.

One approach to overcoming TKI resistance may be to target signaling molecules downstream of FLT3. The PI3K/AKT/mTOR pathway is upregulated in FLT3 inhibitor-resistant AML cells and promotes cell survival and proliferation [11,12]. Combined FLT3 and AKT/mTOR inhibitors act in a synergistic manner [13,14]. AURORA kinase (AURK) inhibitors are also emerging as new agents in the treatment of AML [15]. AURORA kinases A, B, and C are highly conserved serine/threonine kinases that regulate chromosomal alignment and segregation during mitosis and meiosis. Aberrant expression of AURORA kinases is implicated in the genesis of AML: an overexpression of Aurora A and B transcript levels has been reported in freshly isolated leukemia cells [16]. A number of AURK inhibitors have entered early phase clinical trials in patients with AML [17].

We have recently discovered that *FLT3*–ITD AML cells are highly sensitive to palbociclib^18^ (IBRANCE by Pfizer). This CDK4/6 kinase inhibitor is associated with manageable toxicity in clinical trials that enrolled patients with solid tumors and hematological malignancies and has been approved for clinical utility in breast cancer treatment. Inhibitory effects of palbociclib in *FLT3*–ITD-dependent cells were assigned to cell cycle blocking effects and to changes in the CDK6-controlled transcription of *FLT3* and *PIM1* [18]. PIM1 has been described as a well-known oncogenic kinase involved in *FLT3*–ITD-induced cell transformation and is upregulated in TKI-resistant AML cells [19,20,21]. Synergistic cytotoxicity was demonstrated between palbociclib and FLT3 or PIM1 inhibitors in *FLT3–ITD* AML cells [18].

We now extend this list and define AKT and AURORA kinases as CDK6-controlled Achilles’ heels of ITD^+^ and TKD^+^ AML. The cell cycle kinase CDK6 is not only required for *FLT3*–ITD-induced cell proliferation but also for the viability of *FLT3*–TKD-dependent cells. The CDK6 protein acts as a transcriptional regulator of *AKT* and *AURK* in a kinase-dependent manner. Palbociclib administration effectively combines with AKT or AURK inhibitors to kill TKD^+^ and ITD^+^ leukemic cells to a significantly higher degree than any one agent alone. Our data thus provide the basis for the development of synergistic combination therapies—palbociclib being the common denominator for a disease entity where, to date, no real cure exists.

## 2. Results

### 2.1. FLT3 Kinase Domain Mutation Renders Cells Sensitive to the CDK4/6 Inhibitor Palbociclib

In relapsed/refractory AML, the clinical benefit of FLT3 inhibitors has been limited by the rapid generation of resistance mutations, including D835 on the *FLT3*–ITD allele [22]. Similarly, patients with only *FLT3*–D835 mutations respond less frequently to most currently available FLT3–TKI [5,6]. We have recently demonstrated the potent activity of palbociclib in *FLT3*–ITD^+^ AML cells [18]. To investigate whether palbociclib might also be effective in the treatment of the as yet “undruggable” mutant form within the kinase domain at codon D835, we first generated an isogenic cell line model based on the well-established murine Ba/F3 cells (Appendix A). An *FLT3*–ITD alteration was used as positive control [18]. *FLT3*–WT, *FLT3*–ITD, *FLT3*–D835Y, and *FLT3*–ITD–D835Y Ba/F3 cells were IL-3 independent. Palbociclib inhibited the viability of *FLT3*–D835Y^+^ cells in vitro in a dose-dependent manner at significantly lower concentrations than required for any effect on cells bearing wild-type (WT) FLT3 kinase (Figure 1A). No effect was seen on cells with a double-mutated *FLT3*–ITD–D835Y gene.

We next analyzed cell cycle profiles upon palbociclib exposure (Figure 1B and Appendix A). CDK4/6 kinase inhibition caused an accumulation in the subG_1_ compartment, which contains dead cells, only in *FLT3*–D835Y^+^ or *FLT3*–ITD^+^ cells. Palbociclib treatment increased the proportion of *FLT3*–D835Y^+^ cells in subG_1_ from 9.1 ± 0.7% (DMSO control) to 28 ± 9.4%. By contrast, no effects were seen in cells harboring *FLT3*–WT or *FLT3*–ITD–D835Y. The drug-induced subG_1_ accumulation in *FLT3*–ITD- and –D835Y-mutated cells was caused by apoptosis induction. This was revealed by the significant increase in annexin V staining that occurred in a dose-dependent manner already at lower concentrations (Figure 1C and Appendix A).

We also evaluated the effects of palbociclib in an *FLT3*–D835Y^+^ subcutaneous tumor xenograft model. The mice were treated three times per week with palbociclib for 17 days immediately after inoculation with mutant Ba/F3 cells. The drug was well tolerated; no significant loss of body weight or lethality was observed. Palbociclib treatment prevented tumor formation, whereas all mice in the control group developed tumors (Figure 1D). In a parallel experiment, palbociclib treatment was initiated immediately after tumors were palpable to evaluate therapeutic effects on pre-existing tumors. Our findings extend to this setting: tumor progression was significantly attenuated in comparison to the control group (Appendix A). These experiments verified that palbociclib is not only able to inhibit cell growth in vitro but also blocks tumor formation in vivo in the context of *FLT3* kinase domain mutation. 

### 2.2. CDK6 Regulates Expression of AKT and AURORA

Activation of the signaling proteins STAT5, RAS/MAPK, and PI3K/AKT is induced upon *FLT3* mutations. Additionally, an aberrantly increased expression of AURORA kinases is found in human myeloid leukemia cell lines and in patient-derived AML samples [17] (Appendix A). In view of the recently described role of CDK6 as a transcriptional regulator, we performed quantitative polymerase chain reaction studies of a set of genes that had been implicated in these signaling pathways (Figure 2A, Appendix A). Palbociclib treatment reduced the mRNA levels of *AKT* and *AURORA kinases* (*AURK*) in *FLT3–ITD*^+^ and *FLT3*–D835Y^+^ Ba/F3 cells in a dose-dependent manner. Similarly, when *FLT3–ITD* mutant human AML cells lines (MOLM-14, PL-21, and MV4-11) were exposed to palbociclib, the level of *AURK* and *AKT* messenger RNA was significantly decreased in a dose-dependent manner at clinically relevant concentrations. We failed to see comparable effects in cells bearing wild-type *FLT3* (THP-1 and NOMO-1) (Figure 2B and Appendix A). CDK6 ChIP seq analysis using a HA-tagged CDK6 revealed that CDK6 is bound at the promoter sites of the *AURKA*, *AURKB*, *AKT1*, and *AKT2* genes and at the intergenic region of the *AKT3* gene, indicating a direct transcriptional regulatory role of CDK6 (Figure 2C). Our data, thus, support a concept where, in addition to the regulatory effect of CDK6 on *FLT3* and *PIM1* kinases [18], transcriptional control of *AKT* and *AURORA* also contributes to the antileukemic activity of palbociclib in mutant *FLT3*-mediated AML.

When we exposed primary *FLT3*–D835Y^+^ patient-derived AML cells to palbociclib, we could recapitulate our findings obtained with Ba/F3 cells (Figure 3A). Palbociclib treatment reduced cell viability and impaired the ability to form colonies in methylcellulose, paralleled by a pronounced drop in the levels of *AKT* and *AURK* mRNA (Figure 3B–D and Appendix A). These data indicate a benefit of a CDK6-directed therapy for AML patients.

### 2.3. Palbociclib Synergizes with AKT- and with AURORA Kinase Inhibitors

Recent studies have demonstrated that dual AURORA/FLT3 inhibitors have better single-agent efficacy than selective FLT3–TKI against mutant *FLT3*-driven AML [15,23,24,25]. Analysis of data from the Genomics of Drug Sensitivity in Cancer Project database also showed *FLT3*-mutated (point mutations and/or copy number alterations) cells derived from patients with various solid tumors and leukemias as a group to be differentially sensitive to single agents targeting AURORA kinases and AKT (Appendix A). We first investigated the synergy of inhibiting CDK6 and AURK by pairwise drug combination viability assays in *FLT3*–D835Y^+^ Ba/F3 cells. Three-dimensional dose–response surfaces delimited by the single dose–response curves were calculated and analysis of the excess over the Bliss additive synergy revealed a pronounced in vitro synergy between palbociclib and danusertib [26], a pan-AURK inhibitor in phase 2 clinical trials (Figure 4A). The largest deviation from predicted values and thus the highest synergy was found at low nanomolar concentrations, which corresponds to concentrations found in patients (Appendix A). Dose–response experiments showed that combined treatment caused a significant drop in the viability of *FLT3*–ITD^+^ Ba/F3 cells, whereas no effect was seen in cells bearing a wild-type *FLT3* or *FLT3*–ITD–D835Y mutation (Appendix A). The synergistic nature in *FLT3*–D835Y^+^ Ba/F3 cells was further confirmed by combining palbociclib with different AURK inhibitors: tozasertib [27], a pan-AURORA inhibitor in phase 2 clinical trials; alisertib [28], a selective AURORA A inhibitor in phase 3 clinical trials; and CCT137690 [25], a potent pan-AURORA inhibitor (Appendix A). We also investigated the effects of a concomitant targeting CDK6 and AKT pathway. mTOR-directed monotherapy of Ba/F3 cells harboring *FLT3*–D835Y by everolimus [29] had only a marginal effect on cell survival, but the simultaneous inhibition of CDK6 kinase caused a significant drop in viability as well as ERK signaling (Figure 4B and Appendix A). Similar results were obtained when palbociclib was combined with MK-2206 2HCl [30], an allosteric AKT inhibitor currently in phase 2 clinical trials: three-dimensional dose–response surfaces were compared with the predicted values using the Bliss additivity model and revealed strong synergy (Appendix A).

Comparable treatment responses were obtained in *FLT3*–ITD-mutated human AML cells. A pronounced in vitro synergy between palbociclib and AURK inhibitors was seen in the MOLM-14 cell line; the response to monotherapy with AURK inhibitors was stronger when CDK6 kinase activity was simultaneously targeted in these cells (Figure 4C and Appendix A). A strong synergy was further detected when palbociclib was administered together with AKT inhibitors (Figure 4D and Appendix A). Altogether, our data reveal synergistically acting vulnerabilities in AML, with CDK6 being the common denominator (Figure 4E).

## 3. Discussion

Therapy for AML demands effective novel measures. Mutated in a significant portion of AML cases and representing a primary trigger of leukemogenesis, the FLT3 receptor tyrosine kinase is an obvious target for therapeutic drug development. A number of TKI have progressed to clinical trials for *FLT3*-mutated AML. However, despite many years of intensive research, major challenges and hurdles remain for the development of more effective FLT3 inhibitors as therapeutic anti-AML drugs. Mutational resistance in patients under the pressure of TKI therapy has been a serious limitation for FLT3 inhibitors. One of the most frequent *FLT3*–ITD resistance mutations discovered to date is the occurrence of point mutations in the kinase domain. ITD and TKD alterations are mutually exclusive at diagnosis; activating mutations within TKD occur at a rate of 7–10% in AML patients. Despite advances in the development of next generation FLT3–TKI, such as midostaurin [31], many fail to inhibit FLT3 kinase activity when AML cells exhibit an *FLT3*–TKD mutation [5,32]. Thus, there is a pressing need to expand the repertoire of inhibitors available to battle TKI-unresponsive AML clones. We here propose that patients with activating *FLT3*–ITD and –TKD mutations may benefit from CDK6-directed therapy.

CDK6 acts redundantly with CDK4 to promote cell cycle progression. Impaired cell cycle deregulation in hematopoietic malignancies has been linked to a combined deregulation of both kinases through mechanisms such as inactivation of CDK inhibitory proteins and aberrant cyclin D expression. In contrast, *FLT3*–ITD- and –TKD-mediated leukemogenesis involves noncanonical functions of CDK6 not shared by CDK4. The cell cycle kinase CDK6 has been pursued as a genetically validated cancer drug target for a broad spectrum of cancers, including AML. A clinical trial is currently testing the CDK4/6 inhibitor palbociclib in mixed-lineage leukemia-rearranged AML (NCT02310243). CDK6 was reported to inhibit the differentiation of neoplastic cells in a kinase-dependent manner, which reportedly can be exploited as a therapeutic strategy using CDK4/6 kinase inhibitors [33]. Moreover, we recently demonstrated that CDK6 facilitates the cancer cell survival of *FLT3*–ITD^+^ AML cells and that FLT3-targeting TKI and palbociclib act synergistically in these cells [18], The rationale for this combination is the fact that inhibition of CDK6 exerts effects that go beyond cell cycle control: the synergistic effects are mediated by an inhibition of cell cycle progression in combination with the loss of CDK6-mediated transcription of *FLT3* and *PIM1.* As PIM kinases phosphorylate and stabilize FLT3 in vitro [34], the combined treatment disrupts a vicious cycle and feed-forward loop. Our data extend these findings in *FLT3*–TKD-mutated cells; we now demonstrate that CDK6 inhibition is also capable of inducing leukemia cell death in this setting. Upon transfection of Ba/F3 cells with retroviral particles bearing activating *FLT3* mutations, the *FLT3* gene was no longer under the control of an endogenous promoter. Despite the absence of a CDK6-mediated regulation of FLT3 expression, the cells reacted highly sensitively to palbociclib treatment, indicating that CDK6 regulates further important cellular targets required for the viability and expansion of *FLT3*-driven leukemic cells. As such, this cellular model allowed us to unmask *AURK* and *AKT* as additional CDK6-controlled vulnerabilities in AML cells: in fact, CDK6 binds onto the chromatin and facilitates transcription of the oncogenic kinases *AURK* and *AKT* in a kinase-dependent manner. Our conclusion is also supported by data obtained with patient-derived primary AML cells expressing an *FLT3*–TKD mutation; palbociclib caused a pronounced inhibition of cell growth and viability in these cells when tested in colony assay. A xenograft model verified the in vitro effects.

The acquisition of other gene mutations and activation of alternative signaling pathways, including AURORA kinases, PI3K/AKT, RAS/MEK/MAPK, and/or STAT pathways, are seen during continuous treatment with FLT3 inhibitors, which is believed to compensate for the loss of FLT3 activity in terms of survival and growth [11,35]. This proposes that the optimal treatment of AML may require FLT3 inhibition combined with the inhibition of additional signaling pathways. The combined suppression of the CDK4/6 and PI3K/AKT/mTOR pathway already proved to be synergistic in cells derived from T-ALL, malignant pleural mesothelioma, and breast cancer [36,37,38]. We now extend these findings to *FLT3–ITD* or –TKD expressing cells. We identified inhibitors against AURK and PI3K/AKT signaling as synergizing with palbociclib in these settings. Strikingly, cells with resistance-associated *FLT3*–ITD–D835Y mutation remained unresponsive to palbociclib treatment. The activation of CDK6-independent alternative downstream signaling may explain this discrepancy and warrants a whole genome wide in-depth investigation of differences at gene expression level.

In summary, our data underscore that an increasing repertoire of nonredundant CDK6 functions contribute to cancer cell survival in a context-specific manner. Our findings link CDK6 kinase activity to increased apoptosis via the impaired transcriptional regulation of signaling molecules and thus identify CDK6 blockade as a therapeutic strategy. There is rapid translational potential; palbociclib has recently received full approval for use in the treatment of hormone-receptor-positive advanced-stage breast cancer [39,40] and is being clinically evaluated in other cancers. The fact that *CDK6*-deficient mice and mice expressing a kinase-dead *CDK6* allele are viable [41,42,43] is consistent with the manageable toxicity observed in patients. By attacking multiple kinases, including *FLT3*, *PIM1*, *AURK*, and *AKT*, whose functions are not fully overlapping but all crucial for *FLT3*-dependent AML growth and survival, palbociclib may reduce the chances of development of *FLT3* resistance mutations and lead to more durable clinical responses. Hence, palbociclib may show a better clinical toxicity profile and provides a clinically applicable therapeutic window for the design of synergistic combination therapies in AML associated with activating *FLT3* mutations. Based on our findings, we propose targeting CDK6 as a preferable treatment for patients with AML.

## 4. Materials and Methods

### 4.1. Cell Culture

Ba/F3 cells were maintained in RPMI-1640 medium supplemented with 10% fetal calf serum (FCS), 50 µM 2-mercaptoethanol, and 100 ng/mL recombinant mouse IL-3 (R&D Systems, Minneapolis, MN, USA). Human AML cell lines were cultured under standard conditions in RPMI-1640 supplemented with 20% (PL-21 and MV4-11) or 10% (THP-1, NOMO-1 and MOLM14) FCS. PL-21 and MV4-11 cells were a gift of Florian Grebien. THP-1, NOMO-1, and MOLM14 cells were a gift of Stefan Fröhling.

### 4.2. Generation of Ba/F3 Cells Expressing FLT3–ITD, FLT3–D835Y, FLT3–WT, and FLT3–ITD–D835Y

Ba/F3 cells stably expressing wild-type *FLT3* or *FLT3* mutations were generated using pMSCV *FLT3*–WT–GFP and the QuikChange site directed mutagenesis kit XL (Stratagene, Santa Clara, CA, USA). The constructs were confirmed by sequence analysis and transfected into Ba/F3 cells with TurboFect (Thermo Scientific, Waltham, MA, USA) according to the manufacturer’s instructions. Cells were selected for GFP expression using a fluorescence-activated cell sorting analysis (FACS) Aria III cell sorter (BD Biosciences, Franklin Lakes, NJ, USA).

### 4.3. Cell Growth Measurement

Palbociclib was obtained from Pfizer (New York City, NY, USA). Danusertib (PHA-739358), alisertib (MLN8237), tozasertib (VX-680), CCT137690, everolimus (RAD001), MK-2206 2HCl, and ipatasertib (GDC-0068) were purchased from Selleckchem (Houston, TX, USA). For dose–response curves and synergy matrixes, cells were plated in triplicates in 96-well plates. ATP content was measured using CellTiterGlo (Fitchburg, WI, USA) according to the manufacturer’s instructions. IC_50_ determination was performed using GraphPad^®^. The percentage deviation from Bliss independency model [44] was determined via the following formula: Exy = Ex + Ey − (ExEy). E represents the effect on viability of drugs x and y, expressed as a percentage of the maximum effect. Cell cycle profiles were obtained by staining cells with propidium iodide (50 μg/mL) in hypotonic lysis solution (0.1% (*w*/*v*) sodium citrate, 0.1% (*v*/*v*) Triton X-100, 100 μg/mL RNAse) and incubating at 37 °C for 30 min before measurement via FACS.

### 4.4. Apoptosis Measurement

Apoptosis induction upon palbociclib treatment was evaluated by staining cells with AnnexinV eFluor450 (eBioscience, ThermoFisher Scientific, Waltham, MA, USA) and 7-AAD Viability Staining Solution PerCP-Cy5.5 (eBioscience) according to the manufacturer’s instructions, followed by FACS analysis.

### 4.5. Transplantation Studies

Mice were maintained under special pathogen-free (SPF) conditions at the Institute of Pharmacology and Toxicology, University of Veterinary Medicine, Vienna. All procedures were approved by the institutional ethics and animal welfare committee of the University of Veterinary Medicine Vienna (BMWFW-68.205/0112-WF/V/3b/2016 [17 June 2016]; BMWFW-68.205/0093-WF/V/3b/2015 [27 May 2015]) and the national authority, according to §§26ff. of the Animal Experiment Act, Tierversuchsgesetz 2012—TVG 2012. Ba/F3 *FLT3*–D835Y cells were subcutaneously inoculated into both flanks of Rag2^−/−^γc^−/−^ mice. The animals were then randomized to receive palbociclib (15 mg/kg) or vehicle control. The mice were dosed 3× a week starting on day 0 or on day 8 after engraftment, respectively, until terminal workup on day 17.

### 4.6. Quantitative Real-Time PCR

RNA was isolated using Trizol (Invitrogen, Carlsbad, CA, USA) or RNeasy Mini Kit (Qiagen, Venlo, The Netherlands) according to the manufacturer’s instructions. Reverse transcription was performed using iScript cDNA synthesis kit (Bio-Rad, Berkeley, CA, USA). Quantitative real-time PCR was carried out using SsoAdvanced SYBR Green Supermix (Bio-Rad) according to the manufacturer’s protocol. Measurements were related to Rplp0 and Hprt (murine or human as appropriate) as reference genes. The primer sequences will be supplied upon request.

### 4.7. Immunoblotting

Cell extracts were lysed in Laemmli buffer, incubated for 5 min at 95 °C, and sonicated for 15 min at 4 °C. Proteins were resolved by 10% Bis-Trispolyacrylamide gels and transferred to nitrocellulose blotting membranes. Membranes were blocked in 5% BSA (bovine serum albumin) for 1h and probed with the appropriate antibody overnight at 4 °C. The detection of bound antibodies was performed by incubation with horseradish peroxidase-conjugated antirabbit or anti-mouse antibodies at room temperature for 1 h, followed by enhanced chemiluminescence, according to the manufacturer’s protocol (Clarity™ ECL, Bio-Rad). Anti-pERK (Thr202/204, Cell Signaling, Boston, MA, USA), anti-ERK (sc-9102, Santa Cruz, Dallas, TX, USA), and anti-HSC 70 (B-6, Santa Cruz) antibodies were used. The densitometric quantification of signals was done using ImageJ 1.84v software (Wayne Rasband, NIH, USA).

### 4.8. Studies on Primary Patient-Derived Cells

Primary cells were obtained from two *FLT3*–D835Y-positive AML patients after written informed consent was given. Mononuclear cells were isolated using Ficoll. The patients’ characteristics are shown in Figure 3A. Diagnoses were established according to French–American–British (FAB) and World Health Organization (WHO) criteria [45,46,47]. Cells were maintained in RPMI-1640 medium supplemented with 20% serum, 100 ng/mL human recombinant stem cell factor (SCF, Peprotech, Rocky Hill, NJ, USA), 100 ng/mL human recombinant IL-3 (Peprotech), and 300 ng/mL human recombinant FLT3–Ligand (Peprotech). Viability upon incubation with palbociclib was determined via flow cytometry. Defined numbers of cells were seeded in methylcellulose with recombinant cytokines and erythropoietin (MethoCult^TM^ H4434 Catalog #04434, STEMCELL Technologies, Vancouver, Canada) and cultured in the presence or absence of palbociclib. Colonies were counted after 10 days. Falcon^®^ 35 mm not tissue culture-treated Easy-Grip style petri dishes were used (Catalog #351008). The study was approved by the ethics committee of the Medical University of Vienna (Vienna, Austria) under protocol number 224/2006 (26 May 2006), 1063/2018 (20 April 2018), and 1184/2014 (10 July 2014).

### 4.9. Chromatin Immunoprecipitation (ChIP)

ChIP experiments using *BCR/ABL^p185+^* cells with HA-tagged CDK6 were performed in accordance to previously described protocols using antibodies against HA [18,48,49]. For ChIP-Seq analysis [50], sequencing reads were quality controlled by FastQC. Quality filtering and the trimming of reads and adapter removal were done using trimmomatic (version 0.36). Quality filtered reads were mapped against the mouse reference genome (Gencode M13) with bwa-mem (version 0.7.15). Blacklisted regions were removed from the analysis using bedtools subtract (version 2.26.0). Multimapping reads and reads with bad mapping quality were removed by removing reads with a mapping quality below 10 using samtools (version 1.3.1). Peakcalling was performed by MACS2 (version 2.1.0) using default parameters.

### 4.10. Statistical Analysis

Statistical analysis was carried out using a 2-tailed unpaired *t* test. Data are presented as mean values ± S.E.M. and were analyzed by GraphPad^®^. Kaplan-Meier plots were analyzed by the log–rank test. Statistical significance is as follows: * *p* < 0.05, ** *p* < 0.01; *** *p* < 0.001; **** *p* < 0.0001.

## Figures and Tables

**Figure 1 ijms-19-03987-f001:**
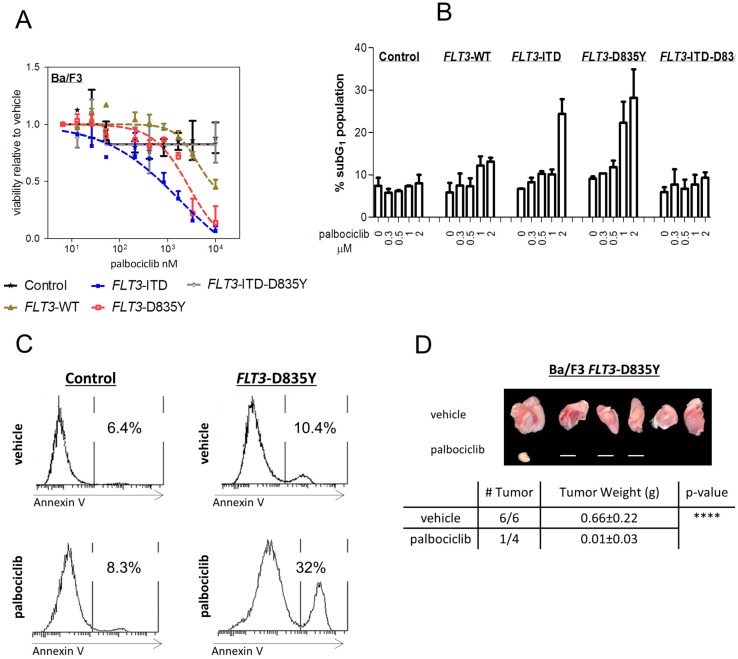
Palbociclib induces apoptosis and impairs *FLT3*–D835Y-driven tumor formation. (**A**) Dose-response curves of Ba/F3 cells transfected with FLT3 variants or empty vector (control) in the presence of palbociclib for 72 h. Cell viability and proliferation were assessed using the Cell-Titer Glo (CTG) assay. (**B**) FLT3 variants were incubated in the absence of cytokines with palbociclib for 72 h, stained with propidium iodide, and analyzed by flow cytometry. Apoptotic subG_1_ fraction is depicted. Error bars indicate ± S.E.M. (**C**) Palbociclib (2 μM)-induced apoptosis was evaluated on day 3 by labeling of indicated cells with annexin V/7-AAD via fluorescence-activated cell sorting analysis (FACS). (**D**) *FLT3*–D835Y^+^ cells were injected subcutaneously into both flanks of immune-compromised Rag2^−/−^γc^−/−^ recipients. Mice were treated 3× a week with vehicle or palbociclib on day 0 (vehicle, *n* = 3 mice; palbociclib, *n* = 2 mice; **** *p* < 0.0001) until terminal workup at day 17. The horizontal line indicates absence of a tumor.

**Figure 2 ijms-19-03987-f002:**
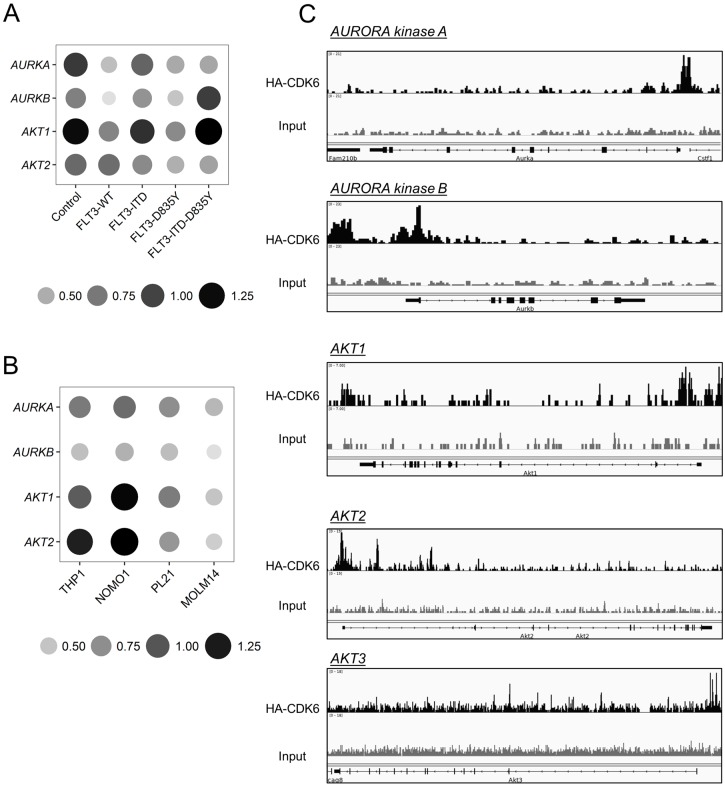
CDK6 binds onto the chromatin of the *AKT* and *AURORA* kinases and regulates their transcription in a kinase-dependent manner. (**A**,**B**) Bubble plot showing relative mRNA levels for indicated genes determined by quantitative RT-PCR when Ba/F3 cells (**A**) and AML cell lines (**B**) were exposed to palbociclib (1 μM (**A**) and 100 nM (**B**)) for 72 h. Relative expression levels were normalized to the housekeeping genes *RPLP0* and *HPRT*. Circle area and color intensity correspond to relative expression. Significance is indicated in Appendix A. (**C**) Representative examples of chromatin immunoprecipitation (ChIP) peaks of indicated *CDK6* targets in murine *BCR/ABL^p185+^*-transformed lymphoid cells expressing HA-tagged CDK6 as a model system.

**Figure 3 ijms-19-03987-f003:**
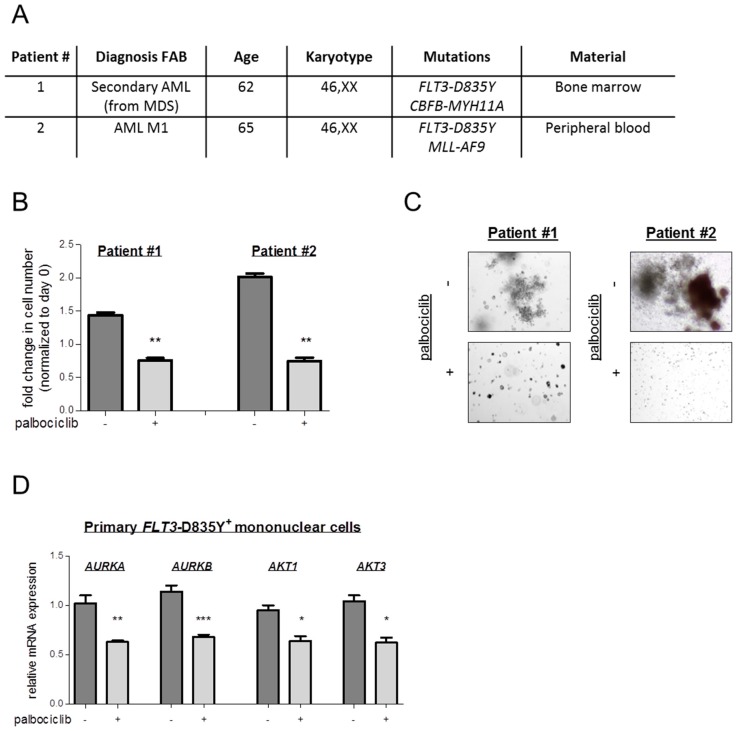
Pharmacologic CDK6 blockade reduces the viability of primary *FLT3*–D835Y^+^ AML cells. (**A**) Characteristics of *FLT3*–D835Y^+^ AML patients. FAB: French–American–British classification; MDS: Myelodysplastic syndrome. (**B**) Primary *FLT3*–D835Y mononuclear cells were incubated in palbociclib (Patient #1: 100 nM; patient 2#: 50 nM). Cell viability was determined by FACS analysis after one week (** *p* < 0.01). (**C**) Patient cells were embedded in methylcellulose with recombinant cytokines and erythropoietin in the absence (−) or presence (+) of palbociclib (patient #1: 100 nM; patient #2: 50 nM). Colonies were counted 10 days after seeding. (**D**) Gene expression was analyzed by quantitative RT-PCR in primary patient specimens after palbociclib treatment (0.3 μM) for 72 h. Relative expression levels were normalized to *RPLP0* mRNA. Error bars indicate ± S.E.M. (* *p* < 0.05; ** *p* < 0.01; *** *p* < 0.001).

**Figure 4 ijms-19-03987-f004:**
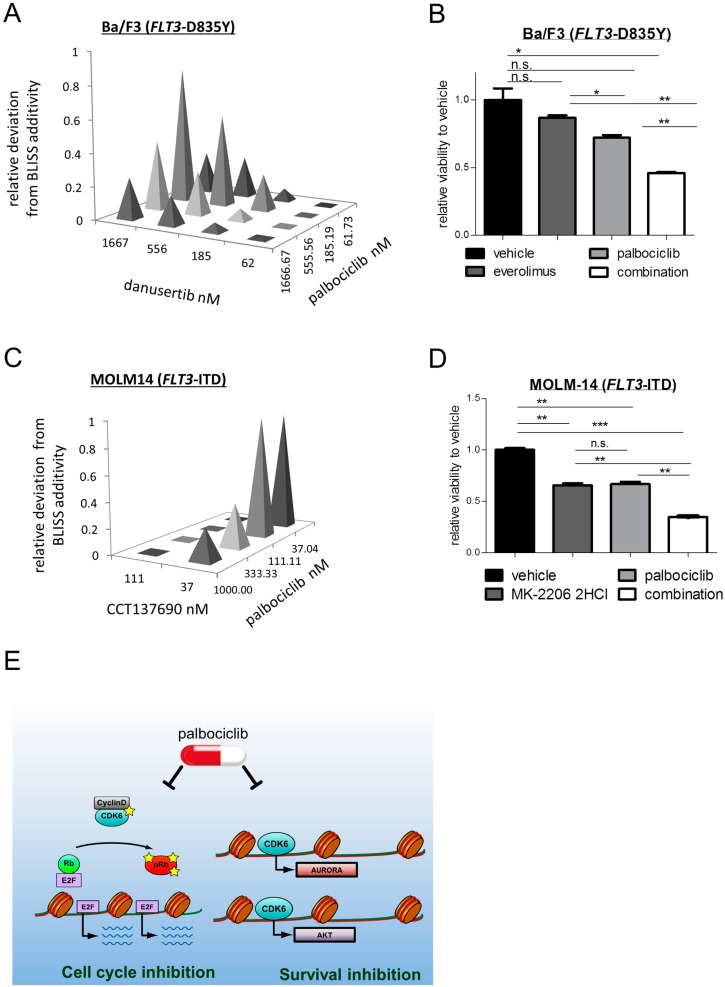
Combined CDK6 and AURK or AKT kinase inhibition exerts synergistic effects. (**A**) Combined effects of palbociclib with the AURORA kinase inhibitor danusertib. Needle graphs indicate deviation from Bliss-predicted additivity in *FLT3*–D835Y^+^ Ba/F3 cells. (**B**) Cells were treated with palbociclib (0.5 μM) and the mTOR inhibitor everolimus (0.1 μM) simultaneously or as single therapy. Cell viability and proliferation was assessed using the CTG assay. Error bars indicate ± S.E.M. (n.s.: Not significant; * *p* < 0.05; ** *p* < 0.01). (**C**) Combined effects of palbociclib with AURORA kinase inhibitor CCT137690. Needle graphs indicate deviation from Bliss-predicted additivity in the *FLT3*–ITD expressing AML cells (MOLM-14). (**D**) Cells were treated with palbociclib (37 nM) and the AKT inhibitor MK-2206 2HCl (37 nM) simultaneously or as single therapy. Cell viability and proliferation was assessed using the CTG assay. Error bars indicate ± S.E.M. (n.s.: Not significant; ** *p* < 0.01; *** *p* < 0.001). (**E**) Graphical summary: CDK6-directed palbociclib treatment impairs cell cycle progression from G_1_ to S phase (left) and unmasks therapeutic vulnerabilities against AURORA and AKT kinase inhibitors in leukemic cells bearing altered *FLT3* (right).

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
