# Peer review of "Therapeutic Vulnerabilities in FLT3-Mutant AML Unmasked by Palbociclib"

_ijms, 2018, doi:10.3390/ijms19123987_

Reviewer 1 Report

In the present study the authors investigate the potential therapeutic validity of the drug palbociclib for treating AML patients with certain mutations. In addition to a previously described role of this drug in inhibiting proliferation and FLT3 transcription in FLT3-ITD+ cells, this study shows an effect of palbociclib on FLT3-TKD+ cells. Their experimental evidence suggests that CDK6 (the target of the drug) regulates the survival of FLT3-TKD+ cells, as well as the expression levels of AKT and AURORA kinases. They show that combined inhibition of CDK6 and AKT/AURORA kinases has a more potent effect on reducing the in vitro survival and growth of cells harbouring FLT3 mutations, compared to single-inhibitor treatments.

The manuscript is well written and the experiments performed correctly, with no major flaws. Even though the synergy between palbociclib and AKT/AURORA inhibitors is shown only in vitro and not in vivo (for example using the Ba/F3 tumour xenograft mouse model), the study provides enough evidence to suggest further investigation of the potential therapeutic use of the drug in combination with existing therapies for AML.

I have a few minor comments that would improve the clarity of the data presented:

1) In Figure 1A, it is not stated how long were the cells treated with the different concentrations of palboclib.

2)For the data presented in Figure 1B and Suppl. Figures 1B and C, a representative FACS plot should be included, showing the gating strategy used to obtain the results. This could be added in the Suppl. Material.

3) In line 130, when describing the experiment presented in Figure 1D, it is stated that mice were treated with the drug daily for 17 days. However, in the figure legend it is written that they were treated 3x per week. One of the two statements should be corrected.

Reviewer 2 Report

Dear Editor,

Thanks for the invitation to review this paper.

Uras et al have investigated the preclinical efficacy potential of using CDK inhibitors in TKD mutant AML. The paper is well written and is the follow up of their recent manuscript describing palbociclib for the treatment of the most recurring mutation in AML which sees an internal duplication of the juxatemembrane domain of FLT3 prognosticating patients into intermediate and high risk categories.

The paper highlights the therapeutic potential of CDK4/6 inhibition in the relapse setting however, the most common and aggressive form of resistance to TKIs is following acquisition of an additional lesion or dual mutation to FLT3, which sees an instigating ITD mutations paired with a kinase domain mutation usually at D835Y/V. The manuscript refers to common resistance forming mutations following the use of TKI, and suggest that these are untargetable, which in part is true however, TKIs that have recently entered routine clinical use such as midostaurin and to a lesser extent crenolanib which are very capable of eradicating FLT3- ITD, -D835V/Y and resistant cells harbouring dual ITD/D835V mutations, however more work is needed for patients harbouring the ITD/D835Y mutation (nice summary DOI:10.3390/ijms19103198)

Using a cell line model they show efficacy in D835Y mutantFLT3 cells, but do not report of D835V mutations / cells? Cells harbouring the dual mutation ITD/D835Y are  however, resistant to palbociclib. WT-FLT3 BaF/3 cells were resistant
to palbociclib. Can the authors comment whether these cells were grown in FLT3 ligand or IL3?

Can the authors also comment why a subcut model was selected for preclinical testing rather than using a bone marrow engraft model, particularly seeing as the cells used in preclinical testing were of mouse origin? What was the assessment of preclinical efficacy resulting from the use of palbociclib in terms of overall survival of the mice?

Can the authors comment on the level of ex vivo annexin V staining following palbociclib?

Given the abundance of data showing that mTOR inhibition drives AKT activity the data showing the combination of palbociclib and everolimus is exciting. What was the effect following the use of these inhibitors alone and then in combination on AKT expression/phosphorylation, and similarly ERK signalling?

The disscussion is to the point and well written.

Author Responses

Round  2

Reviewer 1 Report

The authors have adequately addressed all my comments.

Reviewer 2 Report

Nil